# Investigating traditional and novel predictors of a single versus multiple fragility fractures in a large observational cohort

**Hamzah Amin**[1]*, **Muhammed Aqib Khan**[1], **Marwan Bukhari**[1,2]

**1** Lancaster University Medical School, Lancaster University, Lancaster, United Kingdom, **2** Department of Rheumatology, Royal Lancaster Infirmary, Lancaster, United Kingdom

* h.amin1@lancaster.ac.uk

## Abstract

### Background

Osteoporosis is a systemic skeletal disease characterised by reduced bone mass and a distortion of bone microarchitecture. It is clinically problematic as it leads to fragility fractures which confers excess morbidity and mortality on patients. Up to 32% of individuals will experience recurrent fragility fractures within two years of an initial fracture, yet existing risk models focus on the risk of having a single fragility fracture at a time. We aim to identify predictors of multiple fragility fractures to help improve risk stratification.

### Methods

43,801 patients referred for their first DXA scan in the northwest of England between June 2004 and February 2024 were analysed. Participants underwent lumbar spine and femoral scans to assess bone density and regional body composition. A generalized additive model reporting odds ratios was used to compare risk factors for a single versus multiple fragility fractures.

### Results

Of the referred population, 14,212 (32.4%) had a single fragility fracture and 3,731 (8.5%) had multiple. Female gender was associated with lower odds of multiple fractures (OR 0.88, 95% CI: 0.79–0.99), while increased odds were linked to family history of fractures (OR 1.22, 95% CI: 1.11, 1.35), secondary osteoporosis (OR 1.15, 95% CI: 1.05, 1.26), rheumatoid arthritis (OR = 1.29, 95% CI: 1.08, 1.53), glucocorticoid therapy (OR = 1.18, 95% CI: 1.00, 1.39), smoking (OR 1.27, 95% CI: 1.12, 1.45) and falls risk (OR 2.02, 95% CI: 1.54, 2.63). The combination of falls risk and alcohol consumption increased multiple fracture odds (OR 7.62, 95% CI: 2.77, 20.94). Left femoral T-score and body fat percentage showed significant non-linear effects (both p < 0.001).

**Data availability statement:** Data cannot be shared publicly because of patient confidentiality reasons. Data are available from the NHS trust Institutional Data Access (contact via the research support department: research.support@mbht.nhs.uk) for researchers who meet the criteria for access to confidential data.

**Funding:** The author(s) received no specific funding for this work.

**Competing interests:** Disclosures: HA declares no competing financial interests. MAK declares no competing financial interests MB has the following disclosures: M.B. has been sponsored to attend regional, national and international meetings by UCB Celltech, Roche/Chugai, Pfizer, Abbvie, Merck, Mennarini, Janssen, Bristol-Myers Squib, Novartis and Eli Lilly. He has received honoraria for speaking and attended advisory boards with Bristol-Myers Squib, UCB Celltech, Roche/Chugai, Pfizer, Abbvie, Merck, Mennarini, Sanofi-Aventis, Eli-Lilly, Janssen, Amgen, Novartis and Gilead. He has received honoraria from educational groups Revalidaid and TREG consultants.

## Conclusion

Multiple fragility fractures were associated with many traditional risk factors. We also identified a novel link between falls risk and alcohol consumption, as well as the significant associations with body composition.

## Introduction

Osteoporosis (OP) is a systemic skeletal disease characterised by reduced bone density and microarchitectural distortion of bone [1,2]. Clinically bone density is derived via a dual-energy X-ray absorptiometry (DXA) scan at the lumbar spine and bilateral femurs [3] where the value is compared to a young healthy adult population deriving a T-score. With regards to bone microarchitecture this is derived using advanced algorithms such as the trabecular bone score which analyse lumbar spine DXA scans [4]. OP is a leading cause of fragility fractures which can be defined as fractures resulting in low impact trauma which would not ordinarily result in fracture, including falls from standing height and are associated with increased morbidity and mortality [5,6]. The cost of OP to the United Kingdom's National Health Service is estimated at £4.6 billion, with this figure projected to rise as the population ages [7]. An estimated 20–32% of individuals with a fragility fracture will experience additional fractures within two years of the first fragility fracture, highlighting the importance of identifying patients who will go on to experience multiple fragility fractures [8].

There is a lack of studies specifically examining clinical risk factors for multiple fragility fractures. A notable study by Charles et al. [9] identified several clinical risk factors associated with increased odds of multiple fragility fractures at any site, including age, prior fracture history, history of falls, total hip bone mineral density, spine bone mineral density, and rheumatoid arthritis [9]. In individuals with multiple major osteoporotic fractures, defined as fractures of the femoral neck, vertebrae, proximal humerus, or wrist, the factors associated with increased odds included age, prior fracture history, parental history of hip fracture, total hip bone mineral density, and rheumatoid arthritis [9]. The study also evaluated central fractures, which included fractures of the femoral neck, vertebrae, proximal humerus, pelvis, ribs, scapula, clavicle, or sternum. Predictors of two or more central fractures included age, previous fragility fracture, bone mineral density at the total hip and spine, and higher body mass index, all of which were associated with increased odds of fracture [9]. These findings suggest that while many clinical risk factors are common across fracture types, certain predictors such as body mass index may be site-specific, underscoring the need for tailored risk assessment models in individuals with multiple fragility fractures.

Recently, newer models, such as FRAXplus, have incorporated additional risk factors to update the conventional fracture risk assessments [10]. One of these predictors is the recency of previous osteoporotic fragility fractures, with substantial research suggesting that a prior fragility fracture increases the risk of subsequent fractures [11]. However, it could be argued that identifying individuals at high risk of

multiple fragility fractures before any fracture occurs may offer greater clinical value as early identification would enable timely initiation of both pharmacological and lifestyle interventions, potentially preventing a cascade of fractures. Nevertheless, before predictive models can be applied to distinguish between the risk of a single fracture and multiple fractures prior to any fracture, further research is needed to strengthen the evidence base for predictors specific to multiple fragility fractures.

## Aims

The aim of this study is to investigate the predictors of multiple fragility fractures, evaluating both traditional and novel risk factors, including body composition measurements.

## Methods

### Data collection

43,801 patients were referred from both primary and secondary to our regional NHS DXA clinic for their first DXA scan between June 2004 and February 2024 in northwest England with data accessed on the 13/03/2024 for research purposes. At the time of each scan, trained scanning technicians conducted a structured clinical history, during which patients were asked about the presence of various clinical risk factors for fragility fractures. This information was entered by the technician into a standardised electronic questionnaire integrated within the scanning software, ensuring consistency in data collection.

The questionnaire captured demographic data, including age, as well as clinical risk factors such as family history of fractures, smoking status, current glucocorticoid therapy (defined as ≥5 mg/day of prednisolone or equivalent for ≥3 months), rheumatoid arthritis, excessive alcohol intake (>3 units/day), and known causes of secondary osteoporosis. Secondary osteoporosis was coded for individuals with conditions including polymyalgia rheumatica, coeliac disease, aromatase inhibitor use, amenorrhea, ankylosing spondylitis, anorexia (current or prior), anticonvulsant use (current or prior), breast cancer (current or prior), Depo-Provera use (current or prior), untreated early menopause, hyperparathyroidism (current or prior), hyperthyroidism (current or prior), hypogonadism, inflammatory bowel disease, malabsorption, psoriatic arthritis, systemic lupus erythematosus, and vitamin D deficiency. Data was not available on patients on antiresorptive therapy.

Additionally, patients were asked to self-report any fragility fractures that had occurred within the preceding two years, which was also included in the questionnaire. Fragility fractures were defined as those resulting from a fall from standing height or less, or due to low-impact trauma. All questionnaire responses were cross-checked against patients' medical records to ensure data accuracy and robustness.

### Bone density measurement

The GE Lunar Prodigy system was used for DXA scans between 2004 and 2019, after which the GE Lunar iDXA system was introduced and remains in use. To ensure methodological consistency between devices, phantom scans were conducted at the time of the machine upgrade to validate the use of both machines data in our analysis and in conjunction with ISCD 2019 guidelines [3]. Weekly quality control checks are also performed using phantom scans to maintain measurement accuracy on the DXA scanners [3]. The majority of patients underwent bilateral femoral and lumbar spine scans, with a small subset receiving wrist or isolated femoral scans based where clinically appropriate. Femoral scans included the head, neck, Ward's triangle and the proximal shaft, while lumbar spine scans encompassed the L1 to L4 vertebrae. All scans followed ISCD-recommended patient positioning protocols to optimise image clarity, and exclusion criteria were applied where appropriate as per ISCD guidelines [3], though the reason for exclusion was not explicitly mentioned in the dataset and hence is treated as missing data in our statistical analysis.

### Body composition measurement

Body composition was first assessed by calculating the patients BMI and was conducted by the technician at each appointment. Additionally, DXA scanners can also calculate fat mass and lean mass locally at the time of bone scan, but this data is not reported on the automatically generated bone density reports. Rather this data is stored in the machines database but can be extracted for analysis. Previous studies have outlined the methodology for these measurements [12]. Lean mass and fat mass data from the lumbar spine left and right femoral regions were used to calculate a partial body fat percentage (PBF%). This calculation was based on the sum of fat mass at the spine and bilateral femoral regions, divided by the total fat and lean mass at these sites and multiplied by 100. While this variable has not been clinically validated, previous studies have demonstrated methodological agreement and a strong correlation between regional body composition and total body fat measurements hence the use in this present study [13,14].

### Ethics statement

Ethical approval for pseudonymized data extraction in the absence of informed consent was granted by the Northwest Preston NHS Research Ethics Committee (project number 21/NW/0309). The need for informed consent was waived by the ethics committee. Our data did not include any minors.

### Statistical analysis

All patients who had one or more fragility fractures were included in the statistical analysis. The analysis was conducted using RStudio, with the following R packages: *tidyverse, MGCV, performance, and sjplot.*

Demographic variables were initially compared between the two groups: patients who had a single fragility fracture and those who had multiple fragility fractures. Patients with one fragility fractures were coded as zero, while those with two or more fragility fractures were coded as one. To compare continuous variables, we used the Student's t-test for normally distributed data, and the Mann-Whitney U test was employed for non-normally distributed variables. For categorical variables, the chi-square test was applied to assess differences between groups.

In the primary analysis, we used a generalized additive model (GAM) with a binomial distribution and a logit link function to examine predictors of a single vs multiple fragility fractures. Categorical variables were reported as odds ratios, while continuous variables were fitted using regression splines and reported using marginal effects plot. The GAM automatically selects the optimal smoothing term for continuous variables ensuring we did not under/overfit our data. Variable selection was guided by existing literature and clinical knowledge, rather than automated procedures, to ensure that the most relevant factors were included.

Our model included continuous variables such as age, left femoral T-score, BMI, and PBF%. Categorical variables included rheumatoid arthritis, family history of previous fractures, glucocorticoid therapy, smoking status, excessive alcohol use, fall risk, recurrent falls and secondary osteoporosis. The reference group for these binary variables was the absence of the risk factor. We also included an interaction effect between fall risk/recurrent falls and excessive alcohol use given research suggesting that excessive alcohol consumption may increase both fall risk and the likelihood of recurrent falls [15]. In the context of our interaction effects the reference groups was no for both variables, e.g., falls risk 'no' and excessive alcohol consumption 'no'.

## Results

### Baseline demographics

43,801 patients were referred for their first DXA scan between June 2004 and February 2024. This included 36,480 female and 7,321 male patients. The median age was 66.8 (IQR: 57.6–75.2) years. The median BMI was 26.4 (IQR: 23.2–30.2). 43,801 patients reported 18,037 fractures. Of these fractures, 14,212 (78.7%) patients reported only a single fracture and 3,730 (20.7%) reported 2 or more fragility fractures.

Demographic and clinical characteristics were compared between patients reporting a single fragility fracture (n = 14,212; 79.2%) and those reporting two or more fractures (n = 3,730; 20.8%). There was no significant difference in age or BMI between the groups. However, the multiple fracture group had a slightly lower proportion of female patients, and a higher proportion of male patients compared to the single fracture group (p = 0.033). This group also had a marginally higher PBF%, which was statistically significant. Additionally, they were more likely to be on corticosteroid therapy, have rheumatoid arthritis, consume excessive alcohol, report a family history of fragility fractures, and have a secondary cause of osteoporosis (all p < 0.05). As expected, they also had significantly lower left femoral T-scores (p < 0.001). Full results are presented in Table 1.

We also assessed whether there were differences in risk factors between male and female patients reporting multiple fragility fractures. Male patients were older than female patients (median age: 70 [IQR: 61–77] vs. 66 [IQR: 58–75] years; p < 0.001) and had a slightly higher BMI (27.0 [IQR: 24.1–29.8] vs. 26.1 [IQR: 22.9–30.3] kg/m$^2$; p = 0.007). No other risk factors showed statistically significant differences between males and females reporting multiple fragility fractures.

## Primary analysis

In our primary analysis, we found that females had decreased odds of experiencing multiple fragility fractures compared to males, with an odds ratio (OR) of 0.88 (95% CI: 0.79–0.99; p = 0.029). A family history of fragility fractures was associated with increased odds (OR = 1.22, 95% CI: 1.11–1.35; p < 0.001), as was a diagnosis of rheumatoid arthritis (OR = 1.29, 95% CI: 1.08–1.53; p = 0.004). Glucocorticoid therapy was also significantly associated with higher odds of multiple fragility fractures (OR = 1.18, 95% CI: 1.00–1.39; p = 0.049), as was the presence of a secondary cause of osteoporosis (OR = 1.15, 95% CI: 1.05–1.26; p = 0.002). All results were statistically significant and are presented in Table 2.

**Table 1. Baseline demographics compared between those reporting 1 vs Multiple fragility fractures.**

| | Single Fragility Fracture (n = 14,212, 79.2%) | Multiple Fragility Fracture (n = 3,730, 20.8%) | *P* |
|---|---|---|---|
| Demographics: | | | |
| Age (years) | 67 (58, 75) | 67 (58, 75) | 0.500 |
| Gender | | | 0.033 |
| Male | 2,256 (16%) | 646 (17%) | |
| Female | 11,956 (84%) | 3,084 (83%) | |
| Body Mass Index (kg/m$^2$) | 26.3 (23.3, 30.1) | 26.3 (23.1, 30.1) | 0.400 |
| Lifestyle factors: | | | |
| Alcohol use (>3 units per day) | 368 (2.6%) | 147 (3.9%) | <0.001 |
| Current smoker | 1,371 (9.6%) | 433 (12%) | <0.001 |
| Medical History: | | | |
| Glucocorticoid use | 846 (6.0%) | 271 (7.3%) | 0.003 |
| Rheumatoid arthritis | 738 (5.2%) | 247 (6.6%) | <0.001 |
| Falls risk | 232 (1.6%) | 131 (3.5%) | <0.001 |
| Recurrent falls | 159 (1.1%) | 92 (2.5%) | <0.001 |
| Family history of a fragility fracture | 738 (5.2%) | 247 (6.6%) | <0.001 |
| Secondary cause of osteoporosis | 4,423 (31%) | 1,282 (34%) | <0.001 |
| *DXA* Outcomes: | | | |
| Total Left femoral T-score | −1.30 (−2.09, −0.46) | −1.48 (−2.23, −0.70) | <0.001 |
| PBF% | 31 (26, 36) | 32 (27, 37) | <0.001 |

Data presented as n (%) for binary variable and mean (SD), median (IQR) for continuous variables. Students T-test used for continuous variables and Pearsons chi-squared used for categorical outcomes.

**Table 2. Odds Ratios for the predictors in our GAM looking at predictors of multiple fragility fractures.**

| Risk factor | OR | 95% CI | p-value |
|---|---|---|---|
| Gender | | | |
| Female (reference male) | 0.88 | 0.79, 0.99 | 0.029 |
| Family History of Fracture | 1.22 | 1.11, 1.35 | <0.001 |
| Rheumatoid Arthritis | 1.29 | 1.08, 1.53 | 0.004 |
| Glucocorticoid therapy | 1.18 | 1.00, 1.39 | 0.049 |
| Excess alcohol consumption | 0.98 | 0.83, 1.14 | 0.800 |
| Current Smoker | 1.27 | 1.12, 1.45 | <0.001 |
| Secondary cause of osteoporosis | 1.15 | 1.05, 1.26 | 0.002 |
| Excess alcohol consumption AND Falls Risk | | | |
| Excess alcohol consumption [No] AND Falls Risk [Yes] | 2.02 | 1.54, 2.63 | <0.001 |
| Excess alcohol consumption [Yes] AND Falls Risk [Yes] | 7.62 | 2.77, 20.94 | <0.001 |
| Excess alcohol consumption AND Recurrent Falls | | | |
| Excess alcohol consumption [NO] AND Recurrent Falls [YES] | 2.03 | 1.48, 2.79 | <0.001 |
| Excess alcohol consumption [YES] AND Recurrent Falls [YES] | 3.14 | 1.00, 9.80 | 0.049 |
| Smoothed (PBF%) | | | <0.001 |
| Smoothed (Age) | | | 0.714 |
| Smoothed (Left Femoral Total T-score) | | | <0.001 |
| Smoothed (BMI) | | | 0.450 |

Comparison between reporting 1 vs multiple fragility fractures. The reference category for binary variables was the absence of reporting the risk factor.

Excess alcohol consumption alone was not independently associated with increased odds of multiple fragility fractures (OR = 0.98, 95% CI: 0.83–1.14; $p = 0.756$). In contrast, current smoking was significantly associated with increased odds (OR = 1.27, 95% CI: 1.12–1.45; $p < 0.001$).

We identified significant interactions between alcohol consumption and both falls risk/recurrent falls in relation to reporting multiple fragility fractures. Amongst individuals who did not consume excess alcohol but were identified as a falls risk this was associated with higher odds of reporting multiple fragility fractures (OR 2.02, 95% CI: 1.54–2.63; p < 0.001). The odds increased markedly among those who both consumed excess alcohol and were identified as a falls risk (OR 7.62, 95% CI: 2.77–20.94; p < 0.001). A similar pattern was observed for recurrent falls: in non-drinkers, recurrent falls were associated with increased odds of multiple fractures (OR 2.03, 95% CI: 1.48–2.79; p < 0.001), an increased odds was also seen amongst those who drank excessively and reported recurrent falls (OR 3.14, 95% CI: 1.00–9.81; p = 0.049). These interaction effects are summarised in Table 2.

When examining continuous variables using smoothed terms, both the left femoral total T-score and PBF% demonstrated significant non-linear associations (estimated degrees of freedom [EDF] = 2.99 and 1.01 respectively; $p < 0.001$ for both) with the probability of reporting multiple fragility fractures. In contrast, the smoothed terms for age (p = 0.7) and BMI ($p = 0.4$) were not significantly associated with multiple fragility fractures.

The predicted probability of reporting multiple fragility fractures varied by T-score. A T-score of –4 was associated with a probability of approximately 22.5% to report multiple fragility fractures. This remained relatively flat until a T-score of –2, where the probability declined as bone density improved. This relationship is illustrated in Fig 1.

For PBF%, individuals with 20% PBF% had a predicted probability of approximately 17.5% for reporting multiple fragility fractures, compared to those with 60% PBF%, whose probability rose to approximately 27.5%. This positive association between increasing body fat and the probability of reporting multiple fragility fractures is shown in Fig 2.

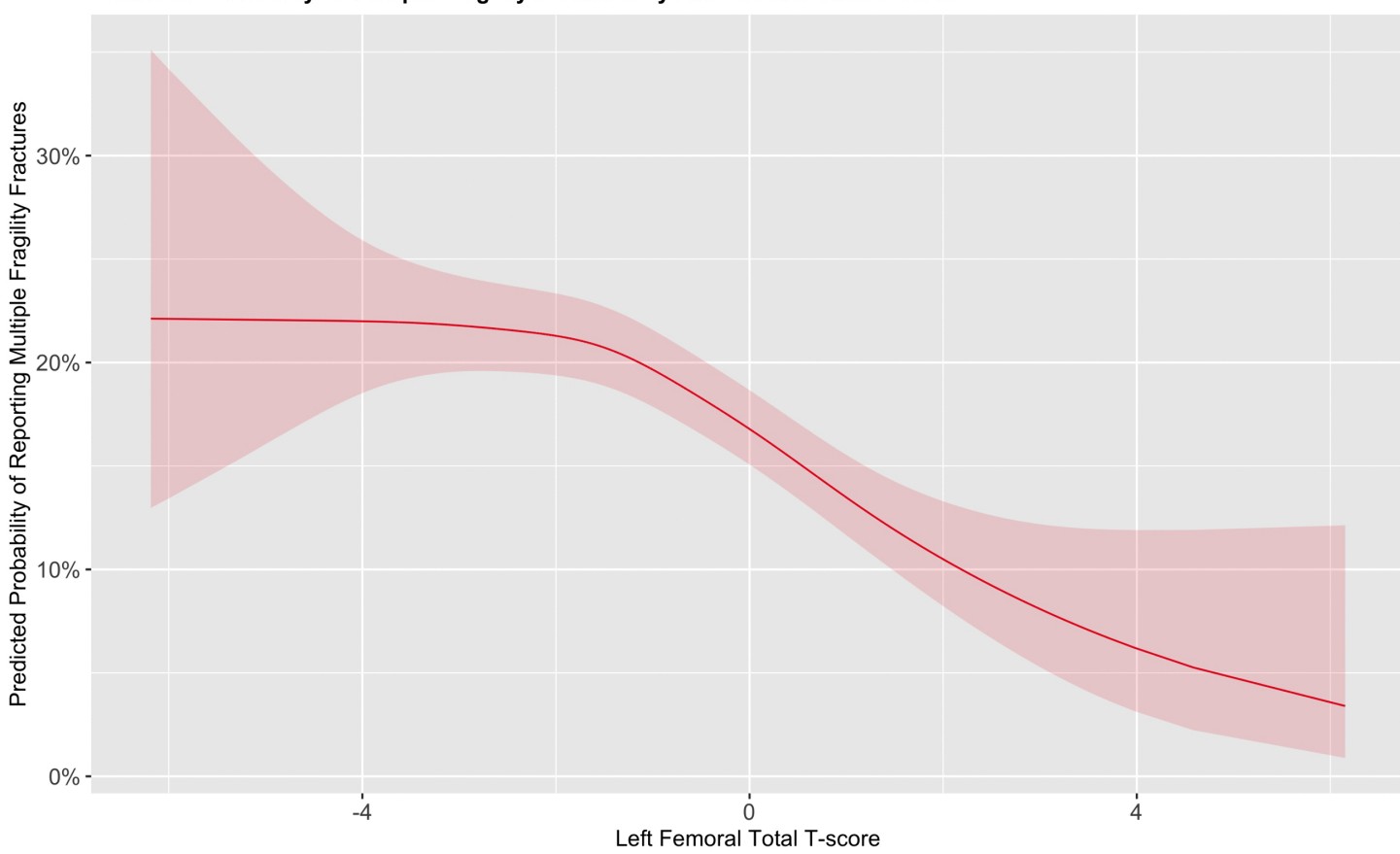

**Predicted Probability of Multiple Fragility Fractures by Left Femoral Total T-score**

**Fig 1. Marginal effects plot for the probability of reporting multiple vs a single fragility fracture based on the left femoral T-score while keeping all other variables in the model constant.**

## Discussion

To our knowledge, this study represents one of the largest clinical observational cohorts to date analysing patients with multiple fragility fractures, building upon the work of Charles et al. [9] and furthering the understanding of factors associated with multiple fragility fractures. We found that 20.8% of patients reported in our cohort reported multiple fragility fractures, which aligns with prior literature suggesting that 20–32% of patients will experience another fragility fracture within two years [8]. This highlights the high risk of experiencing multiple fragility fractures in at-risk populations, emphasizing the need for strategies to identify and treat these patients promptly.

We found that many traditional fractures risk factors were significantly associated with multiple fragility fractures. Interestingly, female gender, typically associated with an increased risk of fragility fractures, was associated with decreased odds in this study. This result should be interpreted with caution, as our sampling approach, which focused on a referred population, likely led to an overrepresentation of females. The males who were referred were generally older and more likely to report multiple fragility fractures, owing to the fact that men are generally less likely to be referred for DXA scans than females [16].

Rheumatoid arthritis was another key factor associated with multiple fragility fractures, increasing the odds by approximately 28% compared to those reporting only a single fragility fracture. Although we lacked access to disease markers,

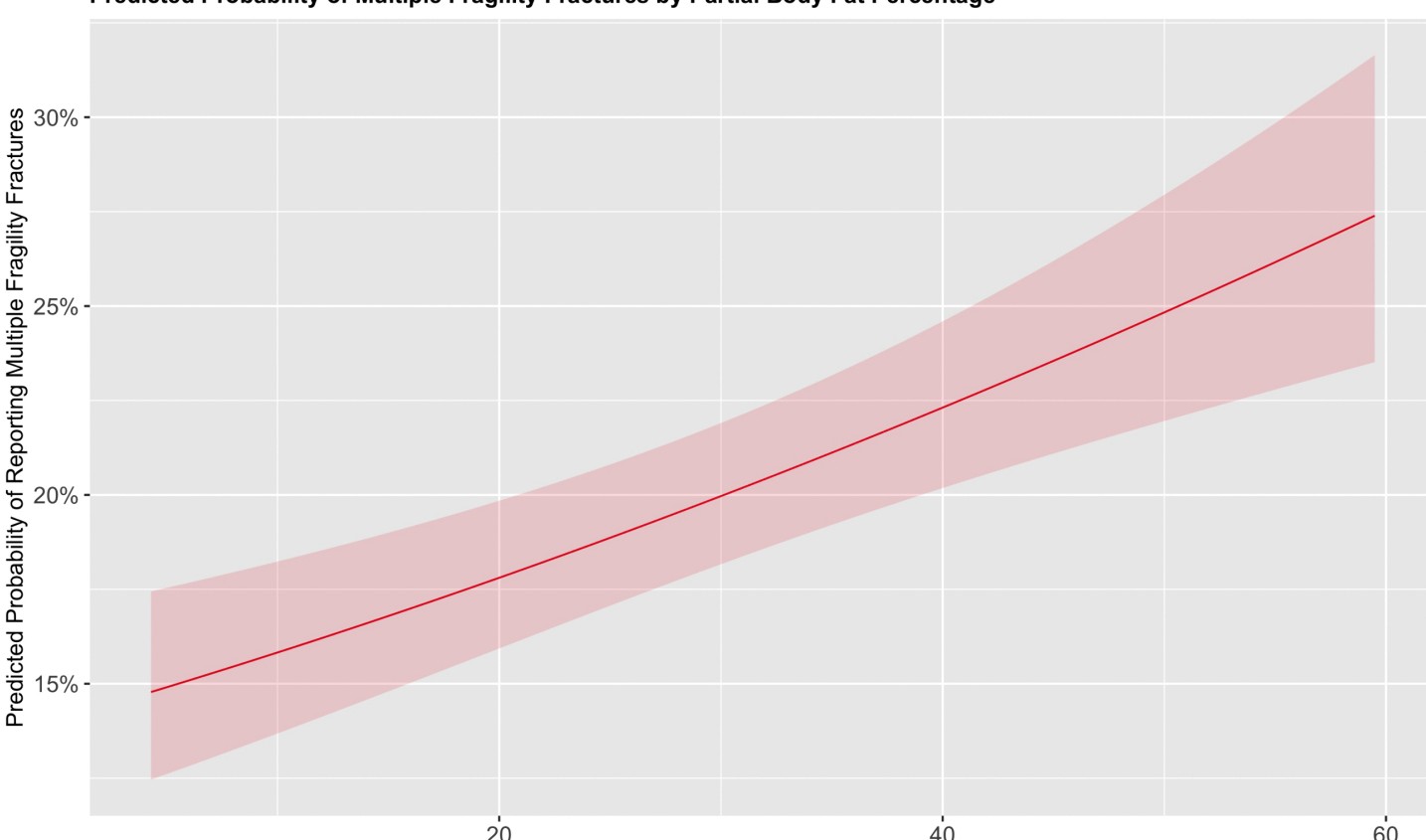

**Predicted Probability of Multiple Fragility Fractures by Partial Body Fat Percentage**

**Fig 2. Marginal effects plot for the probability of reporting multiple vs a single fragility fracture based on the PBF% while keeping all other variables in the model constant.**

previous research has shown that disease activity markers, such as the Disease Activity Score 28 (DAS28), are associated with increased fracture risk [17]. This may be due to the accumulation of disability and joint deformity over time [18], increased falls risk in patients with rheumatoid arthritis [19] as well as the impact on bone density from both disease activity and glucocorticoid use. These findings highlight the importance of effective disease management and the need for clinicians to identify rheumatoid arthritis patients who may be at heightened risk of multiple fragility fractures.

Glucocorticoid therapy was also significantly associated with multiple fragility fractures, increasing the odds by approximately 22%. While the well-established effects of glucocorticoids on bone health are widely recognized [20], the impact on body composition may be overlooked. Glucocorticoid use can induce myopathy and Cushing's syndrome, which is characterized by increased visceral adiposity and muscle wasting [21,22]. These changes can lead to instability and an increased risk of falls [23]. Therefore, while pharmacological treatment to improve bone density is crucial, a holistic approach that addresses compositional changes is also important. Research suggests that exercise may help prevent muscle loss, offering a potential strategy to mitigate the negative effects of glucocorticoid therapy [24].

What also stood out in our statistical analysis was the significant impact of modifiable risk factors on the risk of multiple fragility fractures. Notably, smoking was associated with a 28% increase in the odds of experiencing multiple fragility fractures. Furthermore, While excess alcohol consumption alone did not show a statistically significant

association, we found a noteworthy interaction between alcohol consumption and fall risk. Specifically, patients at risk of falling who did not consume excess alcohol already had a 103% in the increase of odds in reporting multiple fragility fracture versus a single fracture. However, the odds were 683% higher when these patients consumed excess alcohol. This finding aligns with previous research showing that excessive alcohol consumption can impair stability and cognition, ultimately leading to falls especially in the elderly [25]. Hence, given the aging population and evidence suggesting that alcohol use is on the rise, possibly due to factors like social isolation, this is a novel relationship that is not well captured in current fracture risk calculators [26]. However, we acknowledge that the confidence intervals for these interactions were relatively wide indicating uncertainty, likely due to a small number of patients in individual groups. Consequently, larger samples are needed to ascertain the true effect size for drinking alcohol, being at a falls risk and subsequent fracture risk.

We acknowledge the potential criticism regarding the lack of clinical definitions for falls risk and recurrent falls predictors, albeit, this is a common problem in the literature at large [27]. However, we argue that even a clinical suspicion or family concern about a patient's fall risk is sufficient to warrant further investigation. This should include exploring modifiable causes of falls and conducting a comprehensive clinical workup for osteoporosis to prevent fragility fractures. Furthermore, while falls are now starting to be included into FRAXplus [28] it may be wise for an interaction vs additive effect to be included in any risk algorithm to effectively capture this relationship. Hence, further research should explore the correlation between alcohol consumption, fall risk, and fracture risk prediction in prospective cohorts.

Ultimately, alcohol consumption remains a modifiable risk factor which interacts with falls risk to be associated with multiple fragility fractures. Our research suggests that interventions aimed at reducing excessive alcohol intake could be beneficial for patients at risk of multiple fragility fractures by ultimately preventing falls. These interventions may include self-help strategies or therapies such as cognitive-behavioural therapy, which has been shown to effectively reduce alcohol consumption [29]. Furthermore addressing underlying issues, such as social isolation and loneliness, is also crucial and is an ongoing area of research [30]. For patients with severe alcohol dependence, pharmacological treatments may be necessary to reduce alcohol intake however correlation to a reduction in falls risk has yet to be determined.

We also observed that bone density, as measured by the left femoral T-score, was significantly associated with multiple fragility fractures, with nonlinear effects noted. This emphasizes the important of bone density in protecting against multiple fractures [9]. While there are established indications for pharmacological therapies, such as romosozumab, in individuals at imminent risk of fracture, further research should explore the efficacy of these therapies in preventing multiple fragility fractures [31]. Additionally, markers of bone quality such as the trabecular bone score should be explored in relation to multiple fragility fractures, as a proportion of patients may present with normal bone density despite underlying skeletal fragility. Unfortunately, we did not have access to trabecular bone score data in our cohort. Another notable finding from our study is the potential value of body composition measurements as predictors of multiple fragility fractures.

Interestingly, we found that BMI was not significantly associated with reporting multiple fragility fractures. In contrast, our novel PBF% measurement was significantly linked to multiple fragility fractures, with nonlinear effects observed. This further supports the growing evidence that BMI may not be a reliable metric for inclusion in fracture risk calculators given the literature highlighting the importance of additional measures of body composition [32]. However, it is important to recognise that this is not a clinically validated body composition metric, but one derived based on data collected on a routinely collected DXA scan. Hence, we recommend that further research is needed to validate advanced composition model for inclusion in future fracture risk models.

BMI is relatively nonspecific and does not accurately account for the changes in body composition that occur in elderly and at-risk populations for fragility fractures [33]. Elderly patients often experience a reduction in muscle mass and an increase in fat, which research suggests can compromise stability and precipitate falls [34,35]. Our measurements of PBF% may be more sensitive to these changes, as they directly assess a component of compositional change, fat percentage [12]. While some may argue that measuring muscle mass may be more appropriate, much of the literature

suggests that muscle mass is a relatively poor predictor of fragility fractures [36]. This further adds to the evidence supporting fat percentage as a predictor of fragility fractures.

Consequently, novel compositional variables should be considered for inclusion in fracture risk calculators, particularly for patients who have undergone DXA scans. There inclusion could enhance the accuracy and predictive capability of these calculators to predict multiple fragility fracture, facilitating earlier intervention. Furthermore, interventions aimed at improving body composition, such as aerobic exercise and resistance training, should be strongly recommended for this patient population to ensure adequate strength, control, and stability, thereby reducing the risk of multiple fragility fractures [37].

When considering our results in their totality, it may be argued that many of the associations identified in our model are already accounted for within FRAX. However, while FRAX is a useful and widely adopted tool, the dichotomisation of several key variables means changes would be needed to allow it to be useful in predicting multiple fragility fractures. Accounting for a spectrum of alcohol consumption levels [38], glucocorticoid therapy exposure [39], and the number of secondary causes of osteoporosis may improve the ability of fracture risk calculators to discriminate individuals at specific risk of sustaining multiple fragility fractures. Although FRAX values were not available for our cohort, it would also be of interest to assess how FRAX-predicted risk differs between individuals reporting a single versus multiple fragility fractures in a real-world population. This represents an important avenue for future research.

## Strengths and limitations

The strengths of our study include the large sample size of patients with multiple fragility fractures and the comprehensive analysis of comorbidities and risk factors, which enhance the validity of our results. Our results are also strengthened as our patients had the gold standard of DXA for bone density.

However, our study has several limitations. Firstly, our population was predominantly Caucasian (95%+), which may not accurately reflect the broader UK population, although it is representative of the region where the scans were conducted. Additionally, the cross-sectional nature of our study (data collected only at the time of the first scan) limits our ability to draw causal conclusions. Given that our study involved a high-risk referred population, it may not reflect the general population and could be subject to sampling bias. We also acknowledge that recall bias when filling out the questionnaire may have influenced our findings, despite efforts to ensure data accuracy. Furthermore, the lack of specific dosage information for glucocorticoid therapy limits our ability to explore dose-dependent effects on fracture risk, which is crucial for understanding causality. Additionally, the lack of information on patients on antiresorptive represents another limitation as antiresorptive may modify the relationship between patients reporting versus not reporting multiple fragility fractures.

## Conclusions

In conclusion, our study identifies several traditional risk factors for multiple fragility fractures, including family history of fragility fractures, rheumatoid arthritis, glucocorticoid use, smoking, and excessive alcohol consumption. Furthermore, we highlight a novel interaction effect between alcohol consumption, fall risk/recurrent falls in relation to multiple fragility fractures. Additionally, bone density and body composition, particularly PBF%, were found to significantly influence the risk of multiple fragility fractures. Further research is needed to allow accurate risk stratification of patients who will go on to experience multiple fragility fracture prior to any fragility fracture to facilitate timely intervention.

## Author contributions

**Conceptualization:** Hamzah Amin.

**Data curation:** Hamzah Amin.

**Formal analysis:** Hamzah Amin.

**Supervision:** Marwan Bukhari.

**Writing – original draft:** Hamzah Amin, Muhammed Aqib Khan.

**Writing – review & editing:** Hamzah Amin, Muhammed Aqib Khan, Marwan Bukhari.

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
