## [Decision Letter · Decision Letter 0]

7 Jan 2026

Dear Dr. amin,

Thank you for submitting your manuscript to PLOS ONE. After careful consideration, we feel that it has merit but does not fully meet PLOS ONE’s publication criteria as it currently stands. Therefore, we invite you to submit a revised version of the manuscript that addresses the points raised during the review process.

We look forward to receiving your revised manuscript.

Kind regards,

Gaetano Paride Arcidiacono

Academic Editor

PLOS One

**Journal Requirements:**

“Disclosures:

HA declares no competing financial interests.

MAK declares no competing financial interests

MB has the following disclosures: M.B. has been sponsored to attend regional, national and international meetings by UCB Celltech, Roche/Chugai, Pfizer, Abbvie, Merck, Mennarini, Janssen, Bristol-Myers Squib, Novartis and Eli Lilly. He has received honoraria for speaking and attended advisory boards with Bristol-Myers Squib, UCB Celltech, Roche/Chugai, Pfizer, Abbvie, Merck, Mennarini, Sanofi-Aventis, Eli-Lilly, Janssen, Amgen, Novartis and Gilead. He has received honoraria from educational groups Revalidaid and TREG consultants.”

3. In the online submission form you indicate that your data is not available for proprietary reasons and have provided a contact point for accessing this data. Please note that your current contact point is a co-author on this manuscript. According to our Data Policy, the contact point must not be an author on the manuscript and must be an institutional contact, ideally not an individual. Please revise your data statement to a non-author institutional point of contact, such as a data access or ethics committee, and send this to us via return email. Please also include contact information for the third party organization, and please include the full citation of where the data can be found.

**Reviewers' comments** :

Reviewer's Responses to Questions

**Comments to the Author**

1. Is the manuscript technically sound, and do the data support the conclusions?

Reviewer #1: Yes

Reviewer #2: Yes

2. Has the statistical analysis been performed appropriately and rigorously?

Reviewer #1: Yes

Reviewer #2: Yes

3. Have the authors made all data underlying the findings in their manuscript fully available?

Reviewer #1: Yes

Reviewer #2: Yes

4. Is the manuscript presented in an intelligible fashion and written in standard English?

Reviewer #1: Yes

Reviewer #2: Yes

Reviewer #1: OVERALL ASSESSMENT: This is a methodologically sound observational study utilizing a large real-world DXA cohort (n=43,801) to identify predictors of multiple versus single fragility fractures. The GAM analysis is appropriate, confirming traditional risk factors while uncovering a novel alcohol-falls risk interaction (OR 7.60, 95% CI 2.77-20.9) with clear clinical implications. Particularly interesting is the potential role of PBF% as a novel body composition predictor. The study addresses an important gap, as literature specifically examining multiple fracture predictors remains limited.

REQUIRED MINOR REVISIONS: 1. ABSTRACT - BACKGROUND: Please add 1-2 sentences providing brief osteoporosis context, defining it as a chronic systemic disease characterized by reduced bone density, increased fragility fracture risk, and severe clinical consequences (morbidity, mortality, substantial healthcare costs).

2. METHODS - Data Collection: Remove hypothyroidism from the secondary osteoporosis causes list, as the literature does not consistently support it as a fracture risk factor. Indeed, I suggest to remove the patients with hypotyroidism from the secondary osteoporosis group.

3. DISCUSSION - Body Composition: Add a limitation statement clarifying that PBF% represents a partial measure (derived from lumbar spine + bilateral femurs only) rather than validated total body composition analysis. This warrants cautious interpretation of findings and prospective validation in future studies.

MINOR TECHNICAL POINTS

- Table 2: Please clarify the reference category for the alcohol × falls risk interaction terms

- Discussion: Consider briefly acknowledging that the cross-sectional design limits causal inferences

Reviewer #2: The authors have conducted a study addressing an important and clinically relevant question, namely the identification of predictors distinguishing single from multiple fragility fractures in a large observational cohort. The manuscript is generally clear, methodologically sound, and the conclusions are largely supported by the presented data. I believe the study is suitable for publication after minor revisions.

- Introduction. The authors state that osteoporosis is defined by the World Health Organization (WHO) as a reduction in bone density of 2.5 standard deviations below that of a young healthy adult population, measured at the femoral neck. This definition could be reformulated to better reflect the original WHO reference by incorporating both microarchitectural alterations and a more precise densitometric description.

- Results. It may be appropriate to include a study flow chart as the first figure in the Results section to clearly illustrate the number of subjects initially screened and those ultimately included in the final analysis.

- Results. Many of the predictors of multiple fragility fractures identified in this study are already included in FRAX or FRAXplus, which represent established tools for the simultaneous integration of multiple clinical risk factors. It would therefore be of interest to assess whether differences in estimated fracture probability exist between individuals reporting a single versus multiple fragility fractures, and whether a specific probability threshold could be identified to discriminate between these two groups.

- Results. The manuscript does not include information on osteoporosis-specific treatments (e.g. antiresorptive or anabolic therapies), which may substantially influence fracture risk and the occurrence of multiple fragility fractures. The authors should clarify whether treatment data were unavailable or not collected, and acknowledge this as a limitation, as prior or ongoing therapy could confound the observed associations.

- Results. The very high odds ratio observed for the interaction between excessive alcohol consumption and falls risk may be influenced by the relatively small number of individuals in this subgroup. This is also reflected by the wide confidence intervals. The authors are encouraged to acknowledge this limitation explicitly in the Results or Discussion

- General comment. For consistency and clarity, all p-values should be reported to three decimal places throughout the text and tables. When p-values are smaller than this threshold, they should be reported uniformly as p<0.001.

**Do you want your identity to be public for this peer review?** For information about this choice, including consent withdrawal, please see our Privacy Policy

Reviewer #1: No

Reviewer #2: No

---

## [Author Response · Author response to Decision Letter 1]

2 Feb 2026

Please see the attached document.

---

## [Editor Report · Decision Letter 1]

5 Feb 2026

Investigating traditional and novel predictors of a single versus multiple fragility fractures in a large observational cohort

PONE-D-25-39019R1

Dear Dr. amin,

We’re pleased to inform you that your manuscript has been judged scientifically suitable for publication and will be formally accepted for publication once it meets all outstanding technical requirements.

Kind regards,

Gaetano Paride Arcidiacono

Academic Editor

PLOS One

---

## [Editor Report · Acceptance letter]

PONE-D-25-39019R1

PLOS One

Dear Dr. amin,

I'm pleased to inform you that your manuscript has been deemed suitable for publication in PLOS One. Congratulations! Your manuscript is now being handed over to our production team.

Kind regards,

on behalf of

Dr. Gaetano Paride Arcidiacono

Academic Editor

PLOS One